# Comparison between Cardiac Output and Pulmonary Vascular Pressure Measured by Indirect Fick and Thermodilution Methods

**DOI:** 10.3390/jpm13030559

**Published:** 2023-03-20

**Authors:** Igor Volodarsky, Katerina Kerzhner, Dan Haberman, Valeri Cuciuc, Lion Poles, Alex Blatt, Elena Kirzhner, Jacob George, Gera Gandelman

**Affiliations:** 1Heart Center, Kaplan Medical Center, The Hebrew University of Jerusalem, Pasternak St., 1, Rehovot 76100, Israel; woland1978@gmail.com (I.V.);; 2Internal Medicine Department A, Kaplan Medical Center, Pasternak St., 1, Rehovot 76100, Israel

**Keywords:** right heart catheterization, pulmonary hypertension, pulmonary vascular resistance

## Abstract

Introduction: Right heart catheterization (RHC) is a diagnostic procedure, the main purpose of which is to diagnose pulmonary hypertension and investigate its etiology and treatability. In addition to measuring blood pressure in heart chambers, it includes estimating cardiac output (CO) and calculation of pulmonary vascular resistance (PVR) derived from the CO. There are two common methods to evaluate the CO—the indirect Fick method and the thermodilution method. Depending on the clinical conditions, either of the two may be considered better. Several studies have showed that, in most cases, there is no difference between measurements rendered by the two methods. Other studies have raised suspicion of a discrepancy between the two methods in a substantial number of patients. A clear opinion on this matter is missing. Aim: To evaluate the agreement between the values of the CO and PVR found by the thermodilution and indirect Fick methods. Methods: We retrospectively included patients that underwent RHC in Kaplan Medical Center during the last two years with a measurement of the CO using both the thermodilution and the indirect Fick methods. The measurements obtained upon RHC and the clinical data of the patients were collected. The values of the CO and PVR measured or calculated using the two methods were compared for each patient. Results: We included 55 patients that met the inclusion criteria in this study. The mean CO measured by the thermodilution method was 4.94 ± 1.17 L/min and the mean CO measured by the indirect Fick method was 5.82 ± 1.97 L/min. The mean PVR calculated using the thermodilution method was 3.33 ± 3.04 Woods’ units (WU) and the mean PVR calculated using the indirect Fick method was 2.71 ± 2.76 WU. Among the patients with normal mPAP, there was a strong and statistically significant correlation between the PVR values calculated by the two methods (Peasron’s R^2^ = 0.78, *p*-value = 0.004), while among the patients with elevated mPAP, the correlation between the PVR values calculated by the two methods was not statistically significant. Conclusion: The findings of this small study demonstrate that, in a proportion of patients, the indirect Fick method and thermodilution method classify the PVR value differently. In our experience, it seems that, in these patients, the indirect Fick method misclassified patients with a pathological finding as normal. We, therefore, recommend that upon performing RHC, at least in patients with mPAP > 25 mmHg, both the thermodilution and indirect Fick methods be performed and, whenever they disagree, the values obtained from the thermodilution method should be preferred.

## 1. Introduction

Right heart catheterization (RHC) is a well-known procedure, the main indication of which is evaluating pulmonary artery pressure (PAP). It helps to differentiate between different mechanisms of elevated PAP, i.e., pulmonary hypertension, such as primary arterial pulmonary hypertension, pulmonary hypertension related to left heart disease, intrinsic pulmonary disease, pulmonary embolic disease, etc. The main values derived from RHC that help to make the correct diagnosis are the cardiac output (CO) and pulmonary vascular resistance (PVR). The indirect Fick method and the thermodilution method are generally used to this end. Several studies have pointed out that either of these two methods can be more accurate than the other in different clinical settings. For instance, the thermodilution method is known to be inaccurate in low CO states and severe tricuspid regurgitation (TR), though several studies oppose this statement [1,2,3]. As to the accuracy of the indirect Fick method, its main Achilles’ heel is the fact that, according to this method, the CO and PVR are calculated using assumed but not measured oxygen consumption. This method is termed the indirect Fick method as opposed to the direct Fick method, which involves an exact measurement of the oxygen content in inhaled and exhaled air and not estimations based on body size. There are a small number of studies in this field that justify the preference for using the indirect Fick method over the direct Fick method. To make the situation more complicated, the existing studies claim polarly opposed things. On the one hand, there are studies that demonstrate that the thermodilution and the indirect Fick methods give results that do not differ significantly [4,5,6,7,8], whereas studies by Fares and Thrush showed that in a considerable proportion of cases, the values resulting from both methods differ profoundly and may even lead to different clinical decisions [9,10]. Opotowski et al. went as far as to show that discrepancy in estimations of the two approaches can eventually influence the patients’ mortality [11]. Several researchers claim that both measured and assumed oxygen consumption give results that do not diverge considerably from each other, while a more recent study by Grafton et al. pointed out that this does not hold true, for example, in patients who are sicker [12]. This statement especially renders assumed oxygen consumption obsolete, since RHC is commonly indicated in more severely ill patients [12].

Notwithstanding, the Fick method with assumed oxygen consumption (i.e., the indirect Fick method) is still the most widespread method in the majority of catheterization laboratories around the world. This state of affairs is due to several factors. Firstly, the measurement of oxygen consumption requires expensive equipment and trained personnel. Secondly, the thermodilution method is undeservedly deemed to be time-consuming by many operators in the field. Thirdly and, in our opinion, most importantly, is the fact that, disregarding the results of the studies mentioned above, many operators still rely on expert opinions in relation to this subject rather than evidence derived from medical research. Some operators feel that, since most patients with pulmonary hypertension have some degree of TR, and since thermodilution is less reliable in severe TR, a complete discarding of this method is warranted. As a result, these operators share the opinion that whenever the measurement of oxygen consumption is unavailable, using the Fick method with assumed oxygen consumption (indirect Fick method) is the next best thing. In view of this situation, we think that further evidence is needed to show whether there is a significant discrepancy between the indirect Fick method and the thermodilution method, and which one of them is more reliable. In this study, we try to address the question of whether the results of the indirect Fick method can be used and relied on in any clinical setting as is the case currently in most cardiac centers.

## 2. Methods

In this retrospective study, we reviewed records of right heart catheterization performed in our institution from October 2021 backward until the predefined number of patients was reached. Patients who underwent cardiac output and pulmonary vascular resistance using both the thermodilution method and the indirect Fick method were included. When several measurements under different conditions were made for the same patient (for example, at the baseline and after fluid challenge or after iloprost inhalation), only the baseline measurements were used in this study. Patients who underwent RHC combined with additional procedures such as diagnostic or therapeutic coronary arteriography or myocardial biopsy were not excluded, provided that they complied with the aforementioned criteria and all the relevant measurements during RHC were performed. Patients with shunts between the left and right chambers either previously known or demonstrated upon RHC were excluded. Patients with severe tricuspid regurgitation and patients with severe mitral or severe aortic stenoses were excluded from this study also. Following the patients’ selection, we reviewed the medical records of the patients included in the study and collected demographic data, anthropometric data, clinical data, data from echocardiographic imaging and information regarding the relevant diagnostic workup. To comply with the recommendations of the European Society of Cardiologists (ESC) relevant during the time when the patients were evaluated and treated (i.e., not later than October 2021), PVR below 2 Woods’ units (WU) was considered normal and pulmonary hypertension was defined as a mean pulmonary artery pressure of ≥20 mmHg according to the most updated ESC guidelines published in 2022 [13]. However, as the previous version of the same guidelines published in 2015 was still in use at the time the catheterizations were performed and the clinical decisions regarding the patients were taken according to the previous version of the guidelines, we considered the older definitions of the normal ranges wherever it was relevant (PVR below 3, mean PAP < 25 mmHg).

Measurement of the cardiac output using the thermodilution method was performed according to conventional protocols as described in the medical literature [14,15] with the aid of the Horizon HVu System version 3.3a by Mennen Medical Ltd. The indirect Fick method utilized data from oxygen saturation and patient’s hemoglobin levels obtained from the samples drawn during the procedure and calculations were performed using the same system by Mennen Medical Ltd. named above.

The mean and SDs are presented, and a two-sided *t*-test was performed to determine the statistical significance of the differences between the parametrical values. A *p*-value of <0.05 was considered significant. Statistical analysis was performed using IBM SPSS version 21.0 (Armonk, NY, USA).

The study was approved by the Institutional Ethical Board of our Medical Center.

## 3. Results

The medical files of 131 patients who underwent RHC in Kaplan Medical Center from 1 January 2018 to 30 October 2021 were reviewed. In total, 55 patients with a total of 86 series of measurements under different conditions met the inclusion/exclusion criteria. The baseline data of the patients are represented in Table 1. Three patients had atrial fibrillation during the procedure, other patients had sinus rate.

In total, 26 patients (47.3%) had a mean baseline PA pressure of over 25 mmHg and 40 patients (73%) had a mean baseline PAP of over 20 mmHg. A total of 32 patients (58.2%) had a mean PCWP of over 15 mmHg. A total of 13 patients had mPAP, mPCWP and TPG within the range defined as normal by the ESC guidelines [13]. The mean cardiac output (CO) measured using the thermodilution method was 5.0 ± 1.2 L/min and the mean CO measured using the indirect Fick was 5.9 ± 1.0 L/min. The measurements of pressures upon RHC are summarized in Table 2.

The measurements of the cardiac output and PVR for each patient using both methods are shown in Appendix A. The etiology of pulmonary hypertension was diagnosed upon RHC as precapillary in 11 cases (20%), postcapillary in 12 cases (22%) and was deemed of mixed etiology in 10 cases (18%). In 22 patients (40%), the measurements of pressures in the pulmonary artery were considered normal or near-normal at the discretion of both the invasive cardiologist and the attending physician.

A discrepancy between the cardiac output (CO) and pulmonary vascular resistance (PVR) estimated by the indirect Fick and the thermodilution methods was found in a substantial number of patients. In total, 27 patients (49%) had discrepancies of over 10% between the CO estimated by the indirect Fick and the thermodilution method, with the CO measured by thermodilution found to be lower. Six patients (11%) had discrepancies of over 10% between the CO estimated by the indirect Fick and thermodilution methods, with the CO measured by thermodilution found to be higher. The discrepancy is presented in Figure 1 as the ratios of the CO estimated using the two methods for each patient. Pulmonary vascular resistance (PVR) was found <3 WU with the thermodilution method in 34 patients (62%) and with the indirect Fick in 41 patients (75%). While estimating PVR according to the newer ESC guidelines (normal <2 WU), 30 patients (55%) had elevated PVR according to the thermodilution method and 27 (49%) patients had elevated PVR according to the indirect Fick method.

Figure 2 demonstrates that upon calculation of pulmonary vascular resistance (PVR), a good correlation was observed between the values obtained from the indirect Fick method and the thermodilution method (Pearson’s R^2^ = 0.91). According to the older guidelines, seven patients (12%) were classified differently by the two methods; that is, having normal PVR (≤3 WU) according to the Fick method and elevated PVR (>3 WU) according to the thermodilution method. According to the updated guidelines, three patients (5%) were classified differently by the two methods.

The study population was then broken into two groups by the baseline mean PAP and analyzed separately. In total, 13 patients had mPAP within a normal range (≤20 mmHg) and 42 patients had higher-than-normal mPAP (>20 mmHg) (Figure 3A). Among the patients with normal mPAP, the correlation between PVR values calculated by the two methods had a good, statistically significant correlation (Pearson’s R^2^ = 0.90, *p*-value 0.004). Upon evaluation of only the patients with mPAP > 20 mmHg (Figure 3B), the correlation between the two methods was still preserved (Pearson’s R^2^ = 0.93) but without statistical significance. 

Among 29 patients with mPAP < 25 mmHg, only one patient (3.4%) had PVR ≤ 3 WU according to the Fick method and >3 WU according to the thermodilution method, while the remaining 28 had PVR ≤ 3 WU according to both methods. On the other hand, among 26 patients with mPAP > 25 mmHg, there were six patients (23%) who had PVR ≤ 3 WU according to the Fick method and PVR elevated over 3 WU according to the thermodilution method (*p*-value 0.048).

Upon defining PVR according to the current guidelines (normal PVR ≤ 2 WU, and elevated when >2 WU) and setting the mPAP cutoff point at 20 mmHg, among the patients with mPAP ≤ 20 mmHg, all patients had PVR ≤ 2 WU according to both the thermodilution and the indirect Fick methods. Among 42 patients with mPAP over 20 mmHg, only three patients had an incongruent estimation of PVR (≤2 WU according to the thermodilution and >2 WU according to the indirect Fick method), i.e., with the current definitions, the incongruence between the two methods did not cause as much misclassification as with previous definitions, but still was present.

## 4. Discussion

The method of estimating the cardiac output by means of measuring oxygen consumption was introduced in 1870 by German physiologist Adolf Fick, and from that point on it has been considered the gold standard [16]. In 1897, G.N. Stewart described the thermodilution method for the first time in animal research [17] and later in 1971, Ganz et al. [18] demonstrated it for the first time in humans. It is generally accepted that the thermodilution method should not be used in the case of very low cardiac output, severe tricuspid regurgitation, intracardiac shunt or arrhythmia [2,3,15]. However, in less than severe tricuspid regurgitation, this method can be relied on [1]. The thermodilution method is technologically challenging and is dependent on the operator’s familiarity with the technique (i.e., correct timing of the injection at the end of expirium and injecting the correct amount of cold injectate at the correct temperature and at uniform speed).

Currently, the Fick method with the measurement of oxygen consumption is not widely used. Measuring the levels of gases in inspired and expired air is more laborious, more expensive and more time- and energy-consuming than using a mere estimation. It also requires special equipment to measure the gases in inspired and expired air, which most hemodynamic laboratories lack. Thus, the calculation of the cardiac output using the Fick principle based on assumed oxygen consumption is more popular. There are several formulae to estimate oxygen consumption that were proposed and validated by Dehmer et al. in 1970, by LaFarge et al. in 1970 and by Bergstra et al. in 1995 ([19,20,21], Table 3). The Dehmer formula is much simpler; however, it does not take into consideration the patient’s age, gender or heart rate, in contrast to the LaFarge and the Bergstra formulae. All formulae take the patient’s body size into consideration.

Presently, sparse evidence exists in the literature as to which method should be preferred in the absence of the technical contraindications mentioned above (which is the case for the majority of patients undergoing RHC). Some researchers assume that the indirect Fick method gives satisfyingly precise results, and therefore more sophisticated studies are not warranted. However, the indirect Fick method is based on a supposition that oxygen consumption depends only on body size and otherwise does not differ substantially between patients. This supposition cannot hold true for everyone [12,22]. Proponents of the thermodilution method, conversely, believe that this method is better than the indirect Fick method even in presence of severe TR and a low cardiac output [1]. Several researchers have expressed concerns regarding a potential bias that can stem from giving up scrupulous calculations and measurements and relying only on estimations of oxygen consumption instead [21,22,23,24,25]. Some studies, however, showed that both the indirect Fick and thermodilution methods give almost similar results in most patients [4,5,6,7,8]. Most of these studies are rather old, from the 1970s, although there are also some more recent studies. Most of these studies included relatively small populations of patients and compared the mean values of the results of pooled cohorts instead of comparing the results of measurements by both methods for each patient individually. In 2017, in a study in this field with the largest population of patients to this day (a registry with about 15,000 procedures), the thermodilution and indirect Fick methods were shown to differ by more than 20% in more than one third of patients, while cardiac outputs measured by both methods in the entire population did not differ significantly [11], which is very close to the figures found in this study.

The direct Fick method is still considered the gold standard; however, until recently, there was not much evidence that the indirect Fick method renders less accurate results. However, in 2019, Grafton et al. published a very elegant work in which they compared measured oxygen consumption with the values estimated by the Dehmer formula and showed that oxygen consumption clearly depends on the gender and functional status of the patient. According to this study, the Dehmer formula estimated oxygen consumption correctly only in male patients considered in a good functional class but not in other groups [12]. The Dehmer formula, by which oxygen consumption equals 125 mL/min per 1 square meter of BSA, holds true only for healthy male persons and actually gets lower the sicker the patient is. If one uses only the indirect Fick method, it inevitably overestimates the oxygen consumption in patients with high NYHA functional class, meaning the cardiac output will be overestimated and the pulmonary vascular resistance underestimated by up to 25% in a considerable portion of patients.

In our study population, the CO measurements in 38% of patients provided by the thermodilution and the indirect Fick methods differed by more than 20%. It seems that the indirect Fick method has a tendency to produce higher values of cardiac output than those evaluated by thermodilution, and consequently lower values of pulmonary vascular resistance. These results correlate well with the studies published before [9,10,12].

Interestingly, Thrush et al. demonstrated in animals that the direct Fick method, which is considered the gold standard for the evaluation of cardiac output (as was said above), actually gives lower values of the cardiac output than those evaluated using the thermodilution method [9]. In our study and in that of Fares et al., the thermodilution method tended to give lower results for the cardiac output than the indirect Fick method. Coupled with the previous studies, our results give the impression that the indirect Fick method is even less reliable than our study may suggest, since, according to Thrush’s results, thermodilution itself also tends to overestimate the actual magnitude of cardiac output.

Since during the time our patients were evaluated, PVR in the range of 2–3 WU was considered normal, the discrepancy between the two methods significantly affected clinical decisions regarding the patients. To illustrate, in our study, in a patient with the most pronounced discrepancy between the measurements performed using the thermodilution and the indirect Fick methods, PVR was deemed within normal ranges according to the indirect Fick method (2.7 WU) and abnormal according to the thermodilution method (5.2 WU) (see patient number 1 in Appendix A). This patient had pulmonary hypertension of mixed origin but predominantly precapillary (with only slightly elevated PCWP or LVEDP). If only the measurements obtained from the indirect Fick method were considered, this patient should have been diagnosed as suffering from only mild pulmonary hypertension secondary to left heart disease, while according to the thermodilution method, he had PHT of predominantly precapillary etiology and could have benefitted from specific therapy directed against pulmonary hypertension.

It should also be noted that, to the further frustration of the indirect Fick method proponents, all three formulae estimating oxygen consumption use body surface area (BSA) as a value estimating body size. This presents an additional problem to the indirect Fick method since BSA is also calculated and not measured. Several formulae exist to calculate BSA, among them formulae by Gehan, Haycock, Mosteller and DuBois (Table 4). As demonstrated in Figure 4, within the ranges of a stature of 140–200 cm and a weight of 60–100 kg, the results of the first three formulae differ by no more than 10% from one another. However, either underestimation or overestimation of the cardiac output by both BSA and VOx_2_ formulae can lead to mathematical coupling and eventually result in gross bias in the final product, i.e., in pulmonary venous resistance estimation. As the PVR measurement has a direct influence on clinical decisions, the risk of detrimental consequences for patients is obvious.

The lower predictability of oxygen consumption in patients with a low functional status is reasonably explained by existing physiological studies that demonstrated several findings in the tissues of patients with chronic heart failure or chronic pulmonary disease. These changes can be most prominently observed in skeletal muscle. Among these, muscular wasting, change in the number and functionality of mitochondria and paucity of capillaries in skeletal muscles of patients with chronic heart failure can be brought as an example [26,27,28,29,30]. This means that patients who are sicker should have a greater discrepancy between the direct and indirect Fick methods. This observation matches the finding of our study, that the discrepancy between the values derived from the thermodilution and indirect Fick methods is more pronounced among patients who have higher-than-normal mPAP (>20), which, in our opinion, is the most important finding of this study. We showed that among patients with normal mPAP according to the current guidelines (≤20 mmHg), the PVR values calculated by the two methods correlate well with good statistical significance (*p*-value 0.004), but as mPAP rises over its recently redefined cutoff point, the statistically significant correlation is lost (*p*-value 0.08). Thus, the ironic conclusion is that the problem of the discrepancy between the two methods is less relevant in patients without pulmonary hypertension but is even more prominent among patients who actually have PHT, i.e., the patients with a higher chance of benefitting from the RHC survey.

In this study, we did not use the gold standard method (direct Fick method) to evaluate the cardiac output. This point remains the main limitation of this study. We intended to show the situation regarding the evaluation of patients’ hemodynamics in real life, where, as we mentioned, the direct Fick is infrequently used. Therefore, it is impossible to discern sharply between patients in which either the thermodilution method or the indirect Fick method reflects more correct measurements. It is hard to claim whether the thermodilution method is better without rechecking the findings by means of the gold standard method. However, taking into consideration the findings of previous studies, that the indirect Fick method tends to overestimate the cardiac output as compared to the direct Fick method [12,24], and having shown the findings of the current study, in which the cardiac output estimated using the thermodilution method appeared lower than the cardiac output estimated using the indirect Fick method, we can assert that the thermodilution method is, in the majority of cases, closer to the genuine values. The next step in our investigations should include the utilization of the direct Fick method with the measurement of gases in inhaled and exhaled air.

## 5. Conclusions

We can summarize that the direct Fick method has been undeservedly underused due to its technical difficulties and cost (which are not seriously cumbersome), but recent studies warrant revising this attitude. This small study demonstrates that the thermodilution method consistently gives lower estimations of the cardiac output compared to the indirect Fick method, which, in combination with previous studies, leads us to the conclusion that, at least in our experience, the indirect Fick method is more prone to unreliability. Therefore, we recommend that whenever the direct Fick method is unavailable and there is no severe TR and no significant shunt, and especially if upon catheterization of pulmonary artery mean PAP measures over 20 mmHg, at least both the thermodilution and indirect Fick methods should be used.

Novel points demonstrated by this study:The discrepancy between the thermodilution and indirect Fick methods has been demonstrated before. On the other hand, the results of other studies implied that this discrepancy is not clinically relevant. This study, for the first time, demonstrates that this discrepancy is clinically more relevant when mPAP is over 20 mmHg, but may be left unattended when mPAP is normal, thus confirming that the new cutoff point for mPAP has clear implications also on the discrepancy between the thermodilution and indirect Fick methods.

## Figures and Tables

**Figure 1 jpm-13-00559-f001:**
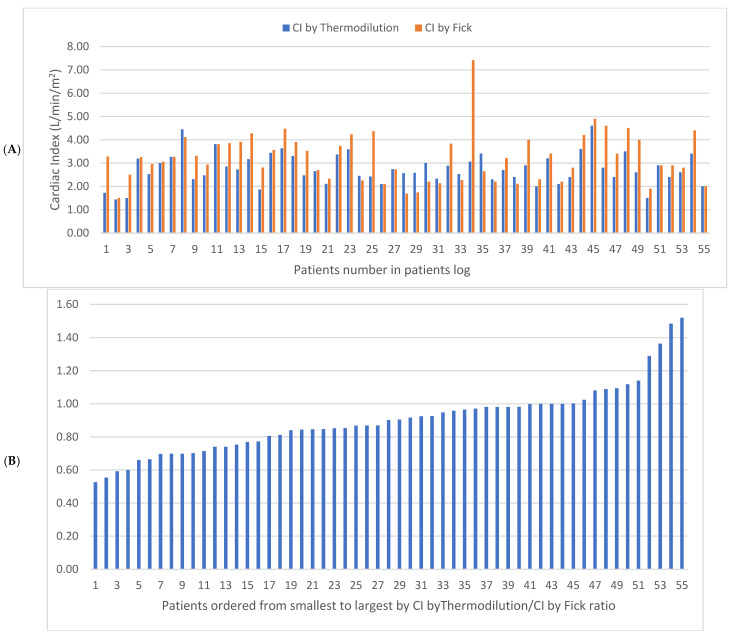
Dataset of 55 patients set in chronological order by date of RHC. (**A**) Cardiac index (CI) by indirect Fick and cardiac index by thermodilution. (**B**) Ratio between the two measurements of the 55 patients. RHC—right heart catheterization.

**Figure 2 jpm-13-00559-f002:**
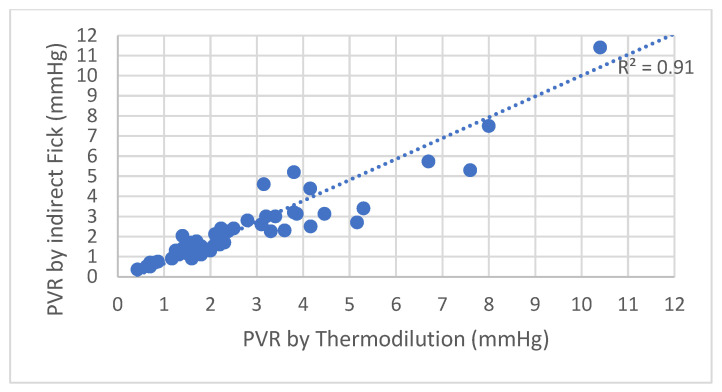
PVR by indirect Fick method plotted against PVR by thermodilution method. PVR—pulmonary vascular resistance.

**Figure 3 jpm-13-00559-f003:**
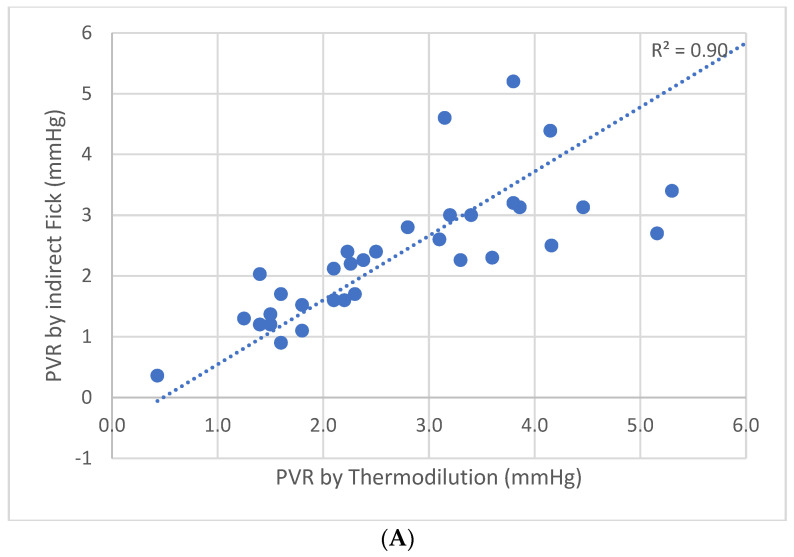
(**A**) PVR using indirect Fick method plotted against PVR using thermodilution method in patients with mean PA pressure > 20 mmHg. (**B**) PVR using indirect Fick method plotted against PVR using thermodilution method in patients with mean PA pressure ≤ 20 mmHg. *p* = 0.004. PVR—pulmonary vascular resistance.

**Figure 4 jpm-13-00559-f004:**
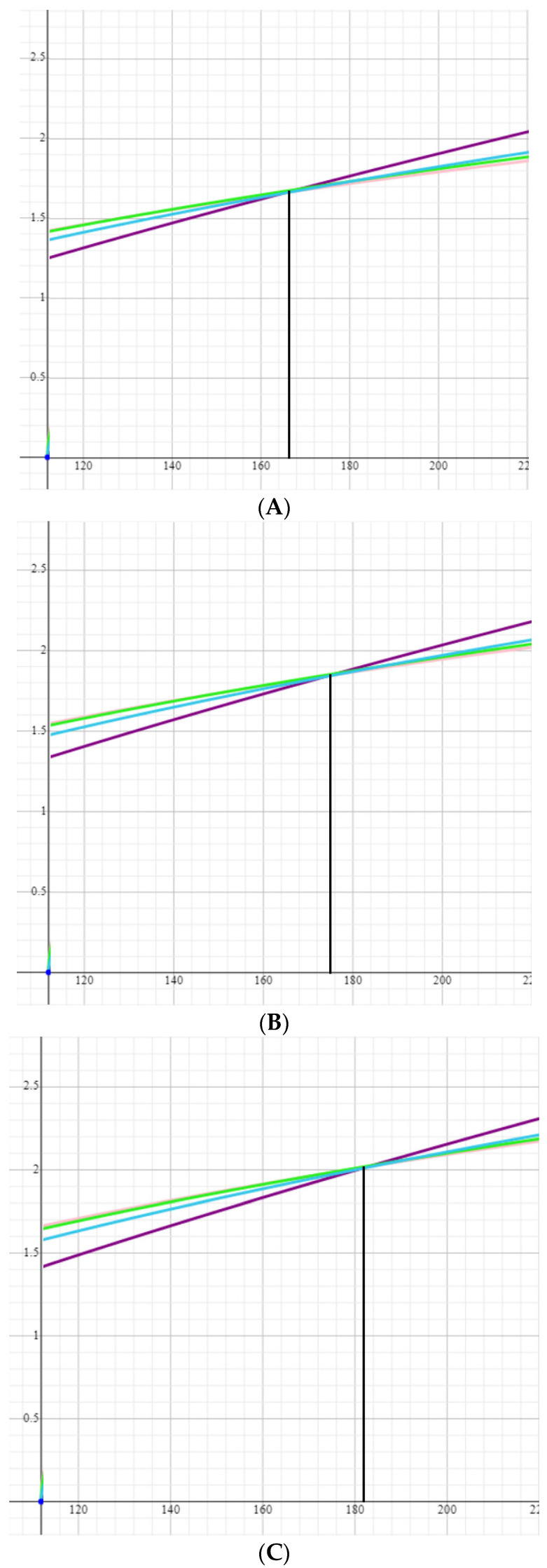
Body surface area by several formulas plotted against stature for persons weighing. (**A**)—60 kg, (**B**)—70 kg, (**C**)—80 kg, (**D**)—90 kg, (**E**)—100 kg. Note that the formulas equalize around stature of 166 cm, 175 cm, 182 cm, 189 cm, 195 cm for persons weighing 60 kg, 70 kg, 80 kg, 90 kg and 100 kg respectively but can differ approximately by 0.1 m^2^, if the actual stature is about 20 cm more or less than the values of equalization as presented above. Legend: 

—Gehan and George formula, 

—Haycock formula, 

—Mosteller formula, 

—Dubois formula.

**Table 1 jpm-13-00559-t001:** Baseline characteristics of the patients.

AGE (YEARS)	69.5 ± 12.5
MALE GENDER	13 (24%)
NYHA FUNCTIONAL CLASS	
I	2 (3.8%)
II	27 (51.9%)
III	18 (34.6%)
IV	5 (9.6%)
COMORBIDITIES	
HYPERTENSION	28 (51%)
DIABETES MELLITUS	13 (24%)
DYSLIPIDEMIA	35 (64%)
OBESITY	14 (25%)
SMOKING	4 (7%)
OBSTRUCTIVE CORONARY ARTERY DISEASE	10 (18%)
PERIPHERAL ARTERY DISEASE INVOLVING CAROTID ARTERY	1 (2%)
PERIPHERAL ARTERY DISEASE INVOLVING OTHER ARTERIES	1 (2%)
S.P. MYOCARDIAL INFARCTION	4 (7%)
ATRIAL FIBRILLATION OR FLUTTER	18 (33%)
S.P. PCI	34 (62%)
S.P. CABG	2 (4%)
CEREBROVASCULAR DISEASE	5 (9%)
S.P. PERIPHERAL EMBOLUS	3 (5%)
THYROID DISEASE	9 (16%)
CHRONIC OBSTRUCTIVE PULMONARY DISEASE	5 (9%)
SIGNIFICANT REGURGITATING VALVULAR HEART DISEASE	9 (17%)
OBSTRUCTIVE SLEEP APNEA	6 (11%)
CHRONIC RENAL DISEASE	32 (60%)
GFR > 90	20 (36%)
GFR 60–90	15 (27%)
GFR 30–90	18 (33%)
GFR < 30	2 (4%)
CONGENITAL HEART DISEASE	1 (2%)
AUTOIMMUNE DISEASE	5 (9%)
VENOUS THROMBOEMBOLIC DISEASE	7 (13%)
CIRRHOSIS	2 (4%)
S.P. MALIGNANCY	3 (5%)
TREATMENT	
ACEI OR ARB	23 (42%)
BETA-BLOCKERS	25 (45%)
MINERALOCORTICOID RECEPTOR ANTAGONISTS	6 (11%)
DIURETICS	22 (40%)
CALCIUM CHANNEL BLOCKERS	12 (22%)
LIPID-LOWERING DRUGS	29 (53%)
ANTIAGGREGANTS	26 (47%)
ANTICOAGULANTS	20 (36%)
HYPOGLYCEMICS	10 (18%)
SPECIFIC TREATMENT AGAINST PULMONARY HYPERTENSION	8 (15%)
PHOSPHODIESTERASE-5 INHIBITORS	4 (50%)
PROSTAGLANDIN AND PROSTACYCLIN ANALOGS	2 (25%)
ENDOTHELIN RECEPTOR ANTAGONISTS	2 (25%)
ECHOCARDIOGRAPHIC PARAMETERS	
LEFT VENTRICULAR EJECTION FRACTION (EF, %)	
EF ≥ 50	51 (93%)
EF 40–50	2 (3.5%)
EF < 40	2 (3.5%)
MEAN THICKNESS (MM)	10.7 ± 1.6
LEFT VENTRICULAR MASS INDEXED (GR/M2)	79.3 ± 37.1
DIASTOLIC DYSFUNCTION GRADE	
IMPAIRED RELAXATION	24 (52%)
PSEUDONORMAL FILLING PATTERN	11 (24%)
RESTRICTIVE FILLING PATTERN	7 (15%)
INCONCLUSIVE	4 (9%)

Legend: NYHA—New York Heart Association, PCI—percutaneous coronary intervention, CABG—coronary artery bypass grafting, GFR—glomerular filtration rate, ACE—angiotensin-converting enzyme, ARB—angiotensin receptor blockers.

**Table 2 jpm-13-00559-t002:** Hemodynamical parameters during the procedure according to catheterization.

Mean Pulmonary Arterial Pressure (mmHg)	
≥25	26 (47%)
<25	29 (53%)
>20	40 (73%)
≤20	15 (27%)
Mean PCWP (mmHg)	
≥15	24 (44%)
<15	31 (56%)
LVEDP (mmHg)	
≥15	22 (46%)
<15	26 (54%)
RVEDP (mmHg)	
≥10	36 (79%)
<10	16 (31%)
Mean right atrial pressure (mmHg)	
≥10	15 (29%)
<10	36 (71%)
Mean systemic arterial pressure (mmHg)	
≥110	11 (25%)
<110	43 (75%)

Legend: PCWP—pulmonary capillary wedge pressure, LVEDP—left ventricular end-diastolic pressure, RVEDP—right ventricular end-diastolic pressure.

**Table 3 jpm-13-00559-t003:** Formulas Used to Calculate Estimated Oxygen Consumption.

Authors, Year	Name	Equation	Cohort Summary
Dehmer, Firth and Hillis, 1970 [19]	Dehmer	VO_2_ = 125 × BSA *	n = 108, mean age 49 years, 64% male, 9% cardiomyopathy
LaFarge and Mettinen, 1970 [21]	LaFarge	VO_2_ = 138.1 − (11.49 × log age) + 0.378 × HR) × BSA * for menVO_2_ = 138.1 − (17.04 × log age) + 0.378 × HR) × BSA * for women	n = 879, mean age not provided, range 3–40 years, 59% male, % cardiomyopathy
Bergstra, van Dijk, Hillege, Lie and Mook, 1995 [20]	Bergstra	VO_2_ = 157.3 × BSA * + 10 − (10.5 × log age) + 4.8 for menVO_2_ = 157.3 × BSA * − (10.5 × log age) + 4.8 for women	n = 250, mean age 34.6 years, 57% male, % cardiomyopathy not provided

Legend: BSA—body surface area, VO_2_—oxygen consumption, HR—heart rate. * BSA calculated with the use of Dubois formula (BSA (m^2^) = 0.20247 × Height (cm)^0.725^ × Weight (kg)^0.0.425^).

**Table 4 jpm-13-00559-t004:** Formulas used to estimate body surface area.

Name of the Formula	Formula
Haycock	BSA (m^2^) = 0.024265 × Height (cm)^0.3964^ × Weight (kg)^0.5378^
DuBois and DuBois	BSA (m^2^) = 0.20247 × Height (cm)^0.725^ × Weight (kg)^0.0.425^
Gehan and George	BSA (m^2^) = 0.0235 × Height (cm)^0.42246^ × Weight (kg)^0.51456^
Mosteller	BSA (m^2^) = ([Height (cm) × Weight (kg)]/3600)^1/2^

BSA—body surface area.

## Data Availability

The data used to perform the study are available in the Appendix A mentioned above.

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
