# Peer review of "Comparison between Cardiac Output and Pulmonary Vascular Pressure Measured by Indirect Fick and Thermodilution Methods"

_jpm, 2023, doi:10.3390/jpm13030559_

Round 1

Reviewer 1 Report

Congratulations for the work done,

the sample size is really very low unfortunately to be a decisive study, therefore I would write less strong conclusions e.g. "in our experience ... so we recommend".

I also recommend citing the new European guidelines of this summer and beyond making a re-evaluation based also on the PAP (PAPs less than or equal to 20), in addition to resistance (less than or equal to 2). 

Reviewer 2 Report

Thank you for the opportunity to review this paper.

This is a retrospective study of a limited number of patients (n=55) that investigate the discrepancy between the results of the measurement ofthe CO and the PVR by the thermodilution method and by the indirect Fick method.

 Despite the small number of patients, the results have some clinical and therapeutic implications  (diagnosis, risk stratification and management of these patients). The definition of PH used in this study is based on the previous ESC Guidelines. The results of the measurement of PVR using the definition of PH from the last ESC Guidelines have some differences (as the author presented on page11, just before the discussion). It is useful an extended presentation of the  results  based on the new Guidelines (meanPAP and PVR) and more comments, respectively, in this direction.   

 Because the meanPAP>20mmHg was implemented in clinical practice since 2022, please include this value in the discussion( the degree of discrepancy between the two methods)  and the recommandations.
